# Validity of Tier 1 Modelling Tools and Impacts on Exposure Assessments within REACH Registrations—ETEAM Project, Validation Studies and Consequences

**DOI:** 10.3390/ijerph17124589

**Published:** 2020-06-26

**Authors:** Urs Schlueter, Martin Tischer

**Affiliations:** BAuA: Federal Institute for Occupational Safety and Health, Unit “Exposure Scenarios”, Friedrich-Henkel-Weg 1-25, 44149 Dortmund, Germany; tischer.martin@baua.bund.de

**Keywords:** exposure models, occupational exposure, REACH, validity, ETEAM

## Abstract

In the last years, the evaluation and validation of exposure modelling tools for inhalation exposure assessment at workplaces received new and highly increased attention by different stakeholders. One important study in this regard is the ETEAM (**E**valuation of **T**ier 1 **E**xposure **A**ssessment **M**odels) project that evaluated exposure assessment tools under the European REACH regulation (Registration, Evaluation, Authorisation and Restriction of Chemicals), (but next to the ETEAM project—as a project publicly funded by the German Federal Institute for Occupational Safety and Health (BAuA)—it is a rather new development that research groups from universities in Europe, but also internationally, investigated this issue. These other studies focused not only on REACH tier 1 tools but also investigated other tools and aspects of tool validity. This paper tries to summarise the major findings of studies that explored the different issues of tool validity by focusing on the scientific outcomes and the exposure on the science community. On the other hand, this publication aims to provide guidance on the choice and use of tools, addressing the needs of tool users. The consequences of different stakeholders under REACH are discussed from the results of the validation studies. The major stakeholders are: (1) REACH registrants or applicants for REACH authorisations, meaning those companies, consortia or associations who are subject to REACH; (2) Evaluating authorities within the scope of REACH, meaning the ECHA (European Chemicals Agency) secretariat and committees, but also the competent authorities of the member states or the European Union; (3) Developers of the different models and tools; (4) Users of the different models and tools.

## 1. Introduction—Regulatory Background of Occupational Exposure Assessments for REACH

REACH requires the registration of chemical substances manufactured in the EU or imported to the EU in amounts of more than 1 tonne per year. If a registered substance is manufactured or imported in quantities of 10 tonnes or more per year, registrants must undertake a chemical safety assessment (CSA) and complete a chemical safety report (CSR). For classified substances, the CSA has to include an exposure assessment for all identified uses. REACH guidance [1] of the European Chemicals Agency (ECHA) names several tools as being suitable for assessing occupational exposure in the substance registration.

For simple screening, tier 1 tools are available (e.g., ECETOC TRA, MEASE, EMKG-EXPO-TOOL) [1] which are designed to easily and quickly differentiate situations as risky or not risky. Therefore, tier 1 tools are designed to give a conservative exposure estimate (i.e., higher than measurements at workplaces) based on a limited number of exposure determinants. More advanced, higher tier tools are mentioned that are expected to give more advanced and accurate estimates of exposure (e.g., STOFFENMANAGER^®^, ART—Advanced REACH Tool, RISKOFDERM) [1].

Where the safety of use is not demonstrated on a tier 1 level, REACH guidance directs to either improvement of Risk Management Measures (RMM) to lower exposure or to higher tier assessments (advanced modelling and/or measurements of exposure). An additional option would be the refinement of the hazard evaluation e.g., through further toxicological testing (this aspect is not dealt with in this publication as it focuses on exposure assessment).

This paper tries to summarise briefly the major outcomes of the validation studies for exposure assessment tools of the last years (with a focus on the ETEAM project) and identify the most common problems of workplace exposure modelling in the REACH framework. Additionally, the authors propose further research and development needs for workplace exposure modelling in chemical regulations, draw conclusions, and give recommendations for several stakeholders in the context of REACH.

Higher tier exposure modelling and the available tools for higher tier modelling are not the focus of this paper and are only touched upon here. Dermal exposure modelling at workplaces is only discussed briefly. Only Marquart et al. [2] published in this regard. However, the ETEAM project made some analyses about dermal exposure aspects during the conceptual analysis and the between-user reliability exercise. It is a common understanding in REACH exposure assessment that the biggest deficiency continues to be the dermal pathway as modelling and measurements for skin exposure are still less advanced than for inhalation exposure. In addition, the quantity of dermal exposure data available was judged to be insufficient to allow for a reasonably comprehensive evaluation of the dermal exposure estimate from the tools. Furthermore, dermal measurements were obtained using different methods, leading to different results. The ETEAM project concluded that the available dermal data did not allow for a meaningful validation of the available dermal modelling tools, like RISKOFDERM or the dermal module of ECETOC TRA.

The issue of model performance and validity has received much attention by different stakeholders, including the scientific community, regulators and the industry [3]. The conclusions and recommendations drawn in this paper are based on all of the studies and take into account all kind of activities that were started, as the view of exposure modelling has become more critical in recent years.

## 2. Results of Validation Studies

### Other (Previous and Current) Validation Studies for Models

The performance of the above-mentioned tools has not been comprehensively evaluated before the ETEAM project, although use of modelling has increased in recent years—especially driven by REACH, but also other factors play a role here—as an alternative to workplace measurements. Only a limited number of studies [4,5,6,7,8,9,10,11] have described and evaluated the performance of these workplace exposure modelling tools (e.g., the EASE model) before the ETEAM project.

In parallel with and after the ETEAM project, the number of validation studies for workplace exposure models and tools increased significantly, and an increasing number of studies dealing with different aspects of model validation and tool performance were published in the scientific literature.

Schinkel et al. [12] investigated the validity of the STOFFENMANAGER^®^ algorithms in a cross-validation study by comparing exposure measurement results with exposure estimates using the STOFFENMANAGER^®^ algorithms. Correlations between observed and predicted exposures, bias and precision were calculated and stratified analyses were performed for scenarios for powder and liquid handling. Based on statistical analyses, the STOFFENMANAGER^®^ algorithm was adapted for one of the scenarios, as the 90th percentile estimate for one out of four algorithms was considered as not conservative enough in this study. The calibration and the cross-validation dataset were then merged into one dataset used for calibrating the adapted STOFFENMANAGER^®^ algorithms. This new calibration resulted in new exposure algorithms for the scenarios evaluated in this study. The authors concluded that the 90th percentile estimates of STOFFENMANAGER^®^ are sufficiently conservative.

A number of case studies were performed by different authors in order to demonstrate the usability of the different tier 1 tools.

Vink et al., for example [13], investigated in 2010 a tiered exposure assessment for the risk characterisation under REACH. They used 1-methoxypropan-2-ol (PGME) as a representative for phase-in substances to be registered under REACH. The aim was developing strategies for hazard identification before resorting to toxicological (in vivo) testing. Greater variability was observed in the exposure estimates from ECETOC TRA or less conservative higher tier models (STOFFENMANAGER^®^; RISKOFDERM), when these results were compared with results from a data-rich approach using measured data. The authors concluded that—when safe use cannot be demonstrated—refinement can be introduced in the estimation of hazard and exposure. Because of the variability of exposure modelling, it may often add more value to invest in realistic exposure data than in toxicity studies.

In two case studies by Kupczewska-Dobecka et al., the ECETOC TRA tool was evaluated. The authors [14] investigated the inhalation exposure to different organic solvents (toluene, ethyl acetate and acetone) at workplaces in Poland for selected process categories (PROC) (The process categories define tasks or process types from the occupational perspective. The PROCs are also differentiated by taking into account the exposure potential for workers during the respective tasks or process types. This descriptor can be assigned to workers’ activities contributing to use. The categories are meant to support harmonised and consistent exposure assessment across sectors and supply chains. The use descriptor included in the description of use is expected to reflect the nature and scope of the activities. The explanations and examples, as presented in the ECHA Guidance R12, should be looked at in order to ensure that the process category assigned is appropriate.) as defined in in the ECHA Guidance R12 [15]. These included PROC 2 (paints and lacquers factory: use in closed, continuous process with occasional controlled exposure), PROC 10 (shoe factory: roller or brush application of glues), and PROC 1 (refinery: use in closed process, no likelihood of exposure). The authors conclude—in agreement with other publications—that the selected PROCs do not precisely describe the situation at the workplaces. For toluene and ethyl acetate, the measured concentrations are within the predicted ranges of ECETOC TRA. For the use of acetone, the tool underestimated the exposure compared to measurements in a shoe factory. 

There is no clear indication that the evaluation of volatile solvents is problematic, since the results that are obtained are inconsistent for different volatile substances (acetone, toluene and ethyl acetate). However, there are indications that the existence of Local Exhaust Ventilation (LEV) plays a role in the quality of the assessment. Additionally, the authors consider the selection of the appropriate PROC as a perquisite of a successful exposure assessment. 

In the second case study [16], the same authors investigated the exposure to Toluene diisocyanate (TDI) and Methylene diphenyl diisocyanate (MDI) during polyurethane foam production in Poland. Again, ECETOC TRA underestimated the exposure level in one case (use of TDI in a PROC 2 scenario, “chemical production or refinery in closed continuous process with occasional controlled exposure or processes with equivalent containment conditions”).

With the aim of model validation, Koppisch et al. [17] used data of the MEGA Exposure Database for an evaluation of STOFFENMANAGER^®^. For two STOFFENMANAGER^®^ model equations (handling of powders/granules and machining) to estimate workers exposure to inhalable dust, measurements were selected from MEGA and grouped in scenarios depending on task, product, and control measures. The model was tested by calculating the relative bias of the single measurements and the correlation between geometric means (GMs) for scenarios. The conservatism was evaluated by checking if the percentage of measurement values above the 90th percentile estimate was ≤10%. For handling, the percentage of measurements with a higher result than the estimated 90th percentile was 11%; for machining, it was 7%. 

The authors concluded that the MEGA database could be used for model validation. However, improvements in the database are necessary for modelling purposes in the future, as frequently relevant contextual information is missing in the datasets. Regarding the use for exposure assessment in the REACH context, the authors concluded that STOFFENMANAGER^®^ can be considered as a useful tier 1 tool because of the relative low bias, the good correlation, and the level of conservatism.

Hofstetter et al. [18] compared ECETOC TRA (v2.0) and ART (v1.0) as REACH-recommended exposure estimation tools and a nearfield, far-field (NF-FF) deterministic model as a purely mathematical model and measured experimental results for a simulated exposure scenario involving the application of a toluene-containing spray paint to a work surface. Air samples were collected to evaluate short-term (15 min) and long-term (240 min) exposures. Eight-hour time weighted averages (TWAs) were calculated and compared with the modelling outputs (ECETOC TRA, ART, NF-FF), resulting in overestimation by a factor of 3.61 for ECETOC TRA, 2.92 for ART and 1.96 for NF-FF. The authors concluded that the tier 1 and 2 tools “performed as expected for the simulated exposure scenario, providing relatively accurate, though conservative, estimates according to the level of detail and precision accounted for in each model.”.

In addition to the above studies, in the last five years, in parallel to the ETEAM project and obviously driven greatly by REACH, a number of working groups investigated the usability of exposure assessment tools for regulatory purposes. The ETEAM project only evaluated tier 1 tools for REACH (maybe with the exception of STOFFENMANAGER^®^ being a model between tier 1 and tier 2) and did not, therefore, include ART [19], which is, at the moment, the only generic/mechanistic higher tier exposure assessment model for inhalation exposure recommended by ECHA. 

The working group of Tinnerberg at the Lund University in Sweden examined over the last years several aspects of model evaluation and different models. In their investigations, they also included ART as a tier 2 model. For reasons of completeness, all publications are referenced here:Between user variability of STOFFENMANAGER^®^ (v5.1) [20];Validity of ART (v1.5) and STOFFENMANAGER^®^ by comparing measurement data from seven different types of industries: wood, printing, foundry, spray painting, flour milling, chemical industry, and plastic moulding industry [21];Evaluation of the usability of ECETOC TRA, STOFFENMANAGER^®^ and ART exposure estimations for occupational safety and health by observation of relevant workplaces [22];Evaluation of the usability of ECETOC TRA (v3.0), STOFFENMANAGER^®^ and ART visiting companies and studying situations at the workplace. The level of protection was calculated for the same exposure situations as for the lack of agreements, but the 90th percentile of the models were used for comparison with the geometric mean of the measurements. ECETOC TRA had the lowest level of protection with 31% of the measured exposure exceeding the modelled exposure, STOFFENMANAGER^®^ with 17% and ART with 3%.

These last findings are published in a doctoral dissertation [23] at the faculty of medicine, Lund University, Sweden.

A different approach was followed in another study by Savic et al. [24]. They recently developed the TREXMO software (TRanslation of EXposure MOdels) which combines several tools: ART v.1.5, STOFFENMANAGER^®^ Version 4.0, ECETOC TRA v.3, MEASE v.1.02.01, EMKG-EXPO-TOOL und EASE v.2.0. This simplifies the application of several tools in order to find, for example, the most conservative assessment.

In the most recent study by Savic et al. [25], ART, STOFFENMANAGER^®^ and ECETOC TRA were compared systematically using the TREXMO tool. The three models’ estimates for 319000 different “in silico exposure situations” were computed and the correlation and consistency between them was investigated. Consistency varied significantly according to different exposure types (e.g., more consistent for vapours than for dusts and solids) or settings (e.g., near-fields were more consistent than far-fields, indoor was better than outdoor exposure).

In the most recent publication [26], the authors developed a new modelling approach that together uses the three most popular models, ART, STOFFENMANAGER^®^ and ECETOC TRAv3, to obtain a unique exposure prediction.

Spinazzè et al. [27] compared the performance of ECETOC TRA, STOFFENMANAGER^®^ and ART regarding accuracy and robustness by comparing available measurement data for exposure to organic solvents and pesticides in occupational exposure situations. ECETOC TRA was not considered acceptable by the authors regarding accuracy, confirming that this model is not appropriate for the evaluation of the selected exposure scenarios for pesticides covering the application of five different pesticide active substances. It is, however, questionable to use ECETOC TRA for pesticide uses as this is not within the scope of this tool. However, STOFFENMANAGER^®^ is considered to produce better estimations. For organic solvents, there were no cases of “strong” underestimation, and all models presented overall acceptable results covering eight different scenarios (e.g., surface cleaning, refueling, boat manufacturing, floor stripping). STOFFENMANAGER^®^ was the most robust model overall.

Ishii et al. [28] performed a case study for the evaluation of the ECETOC TRA model (v3.0) for workplace inhalation exposure to ethylbenzene in Japan. In this study, estimated values of ethylbenzene obtained using ECETOC TRA were compared to measured values, and applicability of ECETOC TRA was studied for Japanese workplaces related to manufacturing (PROC 2, 3, 4, 5, 8a, 8b, 9, 13, 14 or 15) and painting (PROC 7 or 10). Most of the estimated values of manufacturing work were above the measured values, and just three of 52 tasks were below the measured values. However, the estimated values for painting were below the measured values in 27 of 85 tasks. All of the work for which the estimated values were below the measured values consisted of the hull block painting in ship building work, especially for roller painting work (PROC 10).

In a case study, Spee et al. [29] compared REACH Chemical Safety Assessment information with the use of polymethylmethacrylate (PMMA) in floor coating in the Netherlands. Use of PMMA flooring and typical exposure situations during application were discussed with twelve representatives of floor laying companies. Exposure to methyl methacrylate (MMA) during the application of PMMA was measured in the breathing zone of the workers at four construction sites, and 14 full shift samples and 14 task-based samples were taken by personal air sampling. The task-based samples were compared with estimates from ECETOC TRA (PROCs 10 and 19 were assigned by the authors, PROC 5 was mentioned). The task-based measurements, in 12 out of 14 (86%) air samples measured exposure was higher than estimated exposure. Recalculation with a lower ventilation rate and a higher temperature during mixing in comparison with the CSR reduced the number of underestimated exposures to 10 (71%) samples. Estimation with the EMKG-EXPO-TOOL failed to identify unsafe exposure situations for all scenarios, which is in accordance with the measurement outcomes. The results emphasize that ECETOC-TRA exposure estimates in poorly controlled situations need better underpinning.

A critical evaluation about parts of the theoretical background of STOFFENMANAGER^®^ and ART was recently published by Koivisto et al. [30]. The authors raise principal questions about the accuracy of the general ventilation multipliers used and recommend a revision of those parts in both tools. They found that their recalculated general ventilation multipliers were considerably higher and the recalculated NF/FF multipliers were smaller than those used in STOFFENMANAGER^®^ and ART. With these results they concluded that the errors in the general ventilation multipliers should not be ignored and recommended revising the general ventilation multipliers or an integration of the NF/FF model to STOFFENMANAGER^®^ and ART. Some of the tool developers of STOFFENMANAGER^®^ and ART replied to this conclusion recently [31] and contradicted the criticism. However, the tool developers only stated that Koivisto et al. incorrectly presented the differences between their calculations and those in the later paper of Cherrie et al. [32], which were on average about 5% higher, without providing a detailed numerical comparison that would substantiate their criticism. In addition for STOFFENMANAGER^®^, the categorization of parameters and the allocation of scores for categories were partly taken from the work by Cherrie and colleagues [33,34], but were not directly translated into STOFFENMANAGER^®^ as Koivisto et al. [30] assumes. In the view of the authors of this publication, the tool owners have not provided a sufficient explanation that would fully justify their rejection of the criticism. Therefore, this remains an open issue and would need further clarification.

Lee et al. [35] compared ECETOC TRA (v3.1), STOFFENMANAGER^®^ (v7.0) and ART (v1.5) with exposure measurements for organic solvents in Korea, applying similar approaches as in the studies described above. The authors collected measurement values and contextual information extracted from 10 survey reports published by the Ministry of Employment and Labour in the mid-2000s. Using the three tools, seven occupational health professionals predicted inhalation exposure to 10 solvents used for cleaning tasks in 51 situations at 33 companies in 15 industries for PROCs 7, 10 and 13 (as assigned by the authors). As in the other studies above, the authors considered STOFFENMANAGER^®^ to be the most balanced model in terms of good accuracy, high correlation, and medium conservatism in the model predictions. However, ECETOC TRA showed less accurate outcomes and a lower level of conservatism but still had moderate correlations. A systematic tendency to overestimate low exposures and underestimate higher exposures was observed in all models, similar to previous studies.

Lee et al. performed a validation study of the lower [36] and higher-tier models [37] with independent data sets that were measured at a variety of workplaces in the United States. Overall, the EMKG-EXPO-TOOL was found to generate more conservative results than TRAv2 and TRAv3, where TRAv3 appeared to be less conservative than TRAv2. Although the conclusion of this study was limited to the liquids with vapour pressures of more than 10 Pa and only a few PROCs (PROC 5, 10, 13, 15), this study utilized the most transparent contextual information compared to previous studies because measurements were collected for the purpose of model validation, thereby reducing uncertainty from assumptions for unknown input parameters. 

Higher-tier models (STOFFENMANAGER^®^ (v4.5) and ART (v1.5)) were evaluated in terms of accuracy and robustness. For liquids with vapour pressures of more than 10 Pa, STOFFENMANAGER^®^ appeared to be reasonably accurate and robust when predicting exposures. 

A full review and discussion of models’ reliability was published by Spinazzè and Borghi et al. [38] very recently. These authors developed a very comprehensive picture of the current scientific knowledge on model and tool evaluation. In their review they investigated Statistical methods used by different authors;PROCs evaluated or not evaluated for ECETOC TRA, STOFFENMANAGER^®^ and ART in different studies;Substances (i.e., powder and dusty solids, nanopowders, liquids, vapour and mist, volatiles, organic chemicals, petroleum substances, solvents, other substances);Single determinants (i.e., activity and substance emission potential, localized controls, general ventilation multipliers, ventilation rate, room size, amount of aerosol sprayed);Model performance for tier 1 tools (ECETOC TRA, MEASE, EMKG-EXPO-TOOL), higher tier tools (STOFFENMANAGER^®^ and ART) and TREXMO and;Between-user reliability.

Additionally, the authors developed future recommendations for exposure models, summarising the different studies for some of the tools. For ECETOC TRA, an improved version was recommended by adjusting four correction factors for integration of some exposure determinants (i.e., duration, percentage of the substance in the mixture, collective protective measures, and personal protective equipment). For ECETOC TRA, some PROCs were recommended for future consideration (i.e., PROC 10 and 15). In addition, algorithms should be reconsidered for high and medium volatile liquids, professional and industrial use, and situations without local exhaust ventilation. For the EMKG-EXPO-TOOL, estimates for high volatile liquids should be re-evaluated. 

In terms of general harmonisation affecting the different tools, the authors identified the following issues that shall be addressed by the scientific community and the relevant stakeholders: Availability of measurements;Uniformity for data collection and storage in databases;Development of multiple-model approaches and combination with exposure measurements;Harmonization and calibration of input parameters (e.g., room size, ventilation exchange rate, duration, energy, dustiness, …);Harmonization of output and result documentation, and;Improvements in guidance documentation, consensus procedures, training methods, and quality control systems.

The final conclusion by Spinazzè and Borghi et al. [38] is that “selecting which model is the most adequate is challenging.” The main reasons for this are the poor comparability of the different studies (e.g., regarding methodology) used for performance evaluation, scale of the study (e.g., number of measurements for validation) or scope of the investigation (e.g., only limited uses, chemicals, …).

## 3. ETEAM Project

The German Federal Institute for Occupational Safety and Health (BAuA) initiated and sponsored a comprehensive validation project for tier 1 tools for occupational exposure assessments. The Evaluation of the Tier 1 Exposure Assessment Models (ETEAM) project evaluated the following tools, in the version that was available at that time:ECETOC Targeted Risk Assessment v2 and v3 [39], referred to as “ECETOC TRA”;STOFFENMANAGER v4.5 [40], referred to as “STOFFENMANAGER^®^”;EMKG-EXPO-TOOL [41], referred to as “EMKG-EXPO-TOOL” and;MEASE v1.02.01 [42], referred to as “MEASE”.

It is important to note that for “STOFFENMANAGER^®^,” EMKG-EXPO-TOOL and MEASE, new versions were published by the tool developers in the last years. These new versions take into account some of the results of the different validation studies.

The ETEAM project aimed at the comparison of the above-mentioned tier 1 tools (i.e., ECETOC TRA, MEASE, EMKG-Expo-Tool, STOFFENMANAGER^®^) in regard of their conceptual background, inherent uncertainty, scope of application, functionality, user-friendliness, and external validity by comparison with independent workplace measurements.

During the international ETEAM conference hosted in 2014 by BAuA, ECHA [43] and BAuA [44] agreed that almost all exposure estimates provided in REACH registration dossiers are based on generic exposure models, especially ECETOC TRA. Other models are rarely used (and then mostly for refinement). There was also consensus that model users do not always observe the domains of models as foreseen by the model developers.

Although tier 1 models are intended to cover a large number of workplaces, until now, most of the above-mentioned validation exercises were carried out on a limited number of exposure scenarios. Therefore, validation studies still do not cover the broad range of scenarios necessary to provide a comprehensive evaluation of all tools. In addition, it is still not clear whether different tools provide comparable results for the same scenario. Overall, validation is still rather limited, and a comprehensive reliability of the models is not complete. The ETEAM project, as one of the most comprehensive evaluation studies, contributed to all of the above-mentioned questions and provided some possible answers. However, it has to be kept in mind that a lot of open questions remain, and further developments are necessary.

In the volume of the Annals of Work Exposures and Health that also included the most important publications of the ETEAM project [45,46,47], Fransman contributed the editorial [48]. He emphasised that generic exposure assessment tools have given exposure assessors a chance to deal with the enormous burden of risk assessments under REACH. However, the results of the evaluations are worrisome and are considered far from perfect and therefore the tools in question and their outputs need to be interpreted with caution. In order to improve this situation, more knowledge is needed about model functionalities, their applicability domain and their uncertainties. This would be clearly a task for the scientific community. The users of the tools need to be cautious as inappropriate exposure assessments could have serious consequences for human health or companies. Fransman pointed out more tasks for exposure scientists aiming to improve workplace modelling: exposure scientists (or occupational hygienists) are required for an appropriate use of the tools. The use of tools should be supported by the collection of exposure measurements, and these should be used in conjunction with exposure models. For the further development of tools, exposure measurement surveys are needed to increase the insight into exposure variability, to understand the effect of exposure determinants on exposure levels, and to continue the validation and refinement.

## 4. Overview of Methods and Results in the ETEAM Project

Detailed descriptions of the methods and results of the ETEAM project are presented in a number of papers by the scientists who performed the different parts of the ETEAM project (overview [46], between-user reliability [45] and external validation [47]). All the details are included in the original study reports, which are available on the BAuA homepage [49]. Table 1 displays a compilation of study reports and publications of the ETEAM project.

Regarding conceptual evaluation and uncertainty analysis, it can be summarised for the various tools that the underlying concepts, strengths and limitations, sources of uncertainty and effects of determinants show big variances and are difficult to compare in many areas. Therefore, no general conclusion on a “best” tool for inhalation exposure estimations can be made, as this always depends on the situation under evaluation and the user in question. Thus, the most important advice for a potential user regarding model concept is to obtain a maximum of information about his preferred tool and to use it only within its designated domain.

### 4.1. Between User Reliability Exercise—BURE

All of the tools were reported to be easy to learn. The results of the Between User Reliability Exercise (BURE) suggest a discrepancy between the perceptions of learning and using the tools and the consistency of the generated estimates [45,53]. Tool guidance did not seem to be consistently applied by users, and more reliable estimates were not generated by those tools that were perceived to be simpler to learn and use. 

Well described exposure situations (information on the majority of parameters were available to the participants) resulted in higher levels of between-user variation. Additional information does not therefore appear to improve consistency between assessors.

Although variation was observed between choices made for the majority of input parameters in the BURE, differing choices of vaguely defined determinants such as
PROC code/activity descriptor and;dustiness level

impacted most on the exposure estimates. 

Errors in allocation of the LEV and local control parameters have a significant effect on the estimate obtained and were a source of considerable variation in the BURE. 

The combination of a standard reporting method, training of users, tool guidance and sector-specific information would form a comprehensive user support framework, which could in turn be the basis of a quality control scheme. Over time, the feedback that users received would allow them to improve and standardise their exposure assessment performance. 

In combination with the results of the uncertainty analysis, it can be summarised that Decreasing the vagueness of model input parameters;Improving implemented tool guidance;Improving information about the exposure situations, and;Model user trainings
are essential for a successful exposure assessment.

Other authors (e.g., [56,57,58]) arrived at very similar results regarding the between-user variability of results of exposure assessments.

### 4.2. External Validation

The external validation study within the ETEAM project is the most comprehensive evaluation of the performance of REACH exposure tools carried out to date [47,54]. The results show that, although generally conservative, the tools may not always achieve the performance specified in the REACH guidance, i.e., using the 75th or 90th percentile of the exposure distribution for the risk characterisation. In particular, for some specific tasks or activities the tools do not always provide sufficiently conservative model estimations. The influence of specific input parameters of the models is more difficult to understand.

Ongoing development, adjustment and recalibration of the tools with new measurement data are essential to ensure adequate characterisation and control of worker exposure to hazardous substances.

## 5. Discussion of the Major Results of the Validation Studies

The development, distribution and use of simple and conservative screening tools were necessary to facilitate the timely processing of the large numbers of exposure assessments for the REACH registrations. The “automated” use of tier 1 tools for “mass production” of exposure scenarios as performed during the phase-in of substances into the REACH registration can lead to serious underestimations of exposures at workplaces as shown by the BURE and external validation within the ETEAM project. Problematic with this approach is the potential lack of thought, and hence “plausibility check” for reasonable Risk Management Measures (RMM) and operational conditions (OC). These “wrong” exposure estimations may result in unhelpful or misleading RMM. Serious consequences for workers’ health might occur if an exposure scenario is incorrectly assessed as “safe” and necessary RMM at the workplace are not implemented. Overestimation of exposure might as well cause an unnecessary financial burden for a company if an exposure scenario is incorrectly evaluated as “unsafe,” leading to overprotective RMM. These overprotective measures might also be a burden for the workers applying them and might have a negative impact on worker compliance.

With regard to the generic character of tier 1 tools, it has been argued—e.g., during discussions of the advisory board of the ETEAM project—that reliability and accuracy (i.e., safety) may have been sacrificed for the sake of simplicity and transparent design. It has therefore been recognised that the tools will require further and ongoing development and validation in the light of the experience gained during the initial registration processes. This initial registration is now finished and therefore it would be an opportunity to take the ETEAM results into consideration.

Unfortunately, the ETEAM database and other evaluation studies performed so far did not include measurement results for all PROCs, and hence the validation exercise concentrated on those PROC codes which were applicable for the majority of the tools. In addition, most studies were based on historical data where coding of the exposure situations into model parameters (due to lack of contextual information) always introduces uncertainty. In order to minimize these uncertainties and to fill up PROC gaps, field studies that directly aim at model validation would be highly desirable. During the field surveys, contextual information required for each tool’s input parameters could be obtained. In addition, pictures and/or video clips could be taken that can facilitate coding of model parameters significantly in case of doubt (Lee et al. [35,36]).

The analyses of the BURE results suggest that when presented with brief, identical descriptions of exposure situations, user variation in the choice of input parameters can lead to very different results. Systematic variation associated with individual users was minor compared to between-user variation. Whilst there was observed variation between choices made for the majority of parameters in the BURE, it appears that the greatest impact on the resulting estimates arises from differences in the choice of vague PROC codes/activity descriptors and of the dustiness level. Additional situation-related information seems to not assist in reducing variation. However, provision of clearer, sector-specific examples of activity description might have a reducing effect on variation.

In conclusion, the BURE results suggest considerable inconsistency between tier 1 tool users. The exposure estimates generally differed by several orders of magnitude per situation and tool, a range which is likely to outweigh any built-in overestimation or other uncertainties within the tools.

The external validation shows the following general results: Tier 1 tools in most cases tend to overestimate exposure, however it remains unclear which level of conservatism shall be expected and accepted for these tools;Tools may not always achieve the performance specified in the REACH guidance with regard to the percentile of the exposure distribution for the risk characterization;Particularly, some of the tools cannot always provide sufficiently conservative exposure estimations for some specific activities;The influence of specific input parameters of the models is more difficult to understand and differs from tool to tool.

A degree of subjectivity will always be present in any assessment process and the nature of tier 1 exposure assessment tools also lead to a high level of uncertainty in the exposure assessment for principle reasons. However, these given facts need to be taken into account when regulatory risk management shall be based on tier 1 assessments. 

## 6. Conclusions and Impacts of Model Validity for Regulatory Exposure Assessment

Central to the concept of a systematic, informed progression in risk assessment is an iterative process of evaluation, deliberation, data collection, work planning, and communication. All of these steps should focus on deciding whether or not the risk assessment, in its current state, is sufficient to support risk management decisions and if the assessment is determined to be insufficient, whether or not progression to a higher tier of complexity would provide a sufficient benefit to warrant the additional effort.

Different stakeholders are affected by the outcomes of the above-mentioned information about validity of worker exposure assessment tools for regulatory risk assessment in general or the results of the different validation studies specifically. These affected stakeholders are:Model/Tool developers/owners should consider the information for revisions and improvements of the models in question;Authorities (ECHA, European Commission, Member State Competent Authorities) using the models for exposure assessments in regulatory frameworks (e.g., REACH) need to consider the information from validation studies when basing regulatory decisions on modelled exposure value;REACH registrants should identify which registrations and uses are affected by the information from validation studies and react accordingly;Industry associations should identify areas and industry sectors where the development of use maps or other means of harmonisation of exposure and use-related information would be beneficial.

To a different extent, all of the above-mentioned stakeholders indeed already responded and took different actions reacting on the new view on tier 1 tools.

## 7. Consequences Drawn for the Tier 1 Tools

All affected model developers have already discussed how further analysis and discussion of the results during the ETEAM project could lead to an improvement of the model in question.

### 7.1. EMKG-EXPO-TOOL

BAuA as the tool developer of the EMKG-EXPO-TOOL released a new version in July 2018 (https://www.baua.de/EN/Topics/Work-design/Hazardous-substances/REACH-assessment-unit/EMKG-Expo-Tool.html) [41]. The revised and improved software version of the EMKG-EXPO-TOOL was developed to improve user-friendliness by way of providing a comprehensive user guide, built-in help features and an interactive user interface. The newly developed version allows the user to generate a report of the results of the assessment of the exposure scenario.

For a summarised overview of the main features and key information of the EMKG-EXPO-TOOL 2.0, a poster is now available for download [59].

In addition to these activities, BAuA recently initiated a new development (https://www.baua.de/EN/Tasks/Research/Research-projects/f2467.html) [60] of an integrated, modular exposure assessment tool that will improve numerous aspects that were identified as deficient in the EMKG-EXPO-TOOL. During the project, a revision of the user interface is foreseen to address the deficiencies that were identified regarding the between-user reliability. The underlying models of the EMKG-EXPO-TOOL will be recalibrated as some shortages were identified, especially for dusty substances. The description and handling of model parameters (e.g., dustiness) will be improved in order to advance decisions for selecting model parameters. The guidance for the practical implementation of model outcomes for workplace design (e.g., OC, RMM) shall be more fit for purpose.

### 7.2. STOFFENMANAGER^®^

The developers of STOFFENMANAGER^®^ recognised the results of the ETEAM project and other validation exercises and “concluded from these studies that STOFFENMANAGER^®^ is the most balanced, robust and sufficiently conservative tool. With the exception that exposure to low volatiles, released as a result of spraying activities outdoors without local exhaust ventilation (aerosol formation- PROC7 and PROC11), was underestimated” [47,61].

In contrast to other tools, STOFFENMANAGER^®^ is commercially distributed and updated more than once a year. Therefore, the current version is STOFFENMANAGER^®^8, whereas during the ETEAM project STOFFENMANAGER^®^ 4.5 was evaluated. Obviously STOFFENMANAGER^®^ is not only the most balanced tool with regard to the level of conservatism and predictive power for volatile liquids and powders—as recognised by Lamb et al. [45] and also the tool developers—but also the most frequently updated tool.

### 7.3. ECETOC TRA

In reaction to the results of the ETEAM project, the European Centre for Ecotoxicology and Toxicology of Chemicals (ECETOC) installed a working group to examine the ETEAM findings, assess the ETEAM database, and identify possible areas of improvements to TRA worker exposure estimates. The major conclusions of this working group [62,63] were that Many instances were found where datasets have only few samples and are not sufficiently robust to draw critical conclusions. ◦A few datasets appear to heavily weigh on overall findings;◦No “statistical test” was applied to determine if the datasets can be considered representative.The ETEAM database contains cases where REACH Use Descriptors were incorrectly applied.
◦This materially affects the nature of the associated ETEAM findings;◦Use map resources can aid ETEAM in assigning proper TRA parameters.Based on the preliminary findings presented above, they suggest ETEAM results on TRA underestimation of some PROCs is premature and less severe than reported by ETEAM.Conclusions in the ETEAM report need further examination.

Further to this analysis, ECETOC together with CEFIC LRI started research projects in order to improve the scientific background of ECETOC TRA. The project “Development of an integrated risk management measure library” aims at an improved and harmonised collection of risk management options for regulatory risk assessments [64]. The project “Experimental assessment of inhalation and dermal exposure to chemicals during industrial and professional activities” aims at the generation of inhalation and dermal exposure data that can be used to evaluate ECETOC TRA worker exposure estimates for key industrial and professional uses of chemicals. Activities are especially in the focus, where in previous validation studies the PROCs were either not investigated (due to insufficient data available) or where the ECETOC TRA underestimates the exposure compared to the measured exposure [65].

Taking into account all the above-mentioned information, ECETOC recently started a systematic review of worker inhalation exposure estimates of the TRA tool [66]. However, results are not yet available, and the implementation will take some time.

### 7.4. MEASE

MEASE is an independent software application that was programmed in Java and will continue to be available free of charge via the EBRC homepage https://www.ebrc.de/tools/downloads.php) [42]. An update of MEASE has been announced by the model developers, but is not available on the EBRC website yet [67]. The model developers tried to address the ETEAM results and believe that a robust exposure assessment tool is now available. 

In particular, they have taken up the ETEAM results regarding the “between-user variability” as well as the “external validation” [67]. The most important points to mention here are the embedding of a “PROC Selection Guide” to reduce between-user variability. An advanced guidance for the selection of physical shape/dustiness and for hot metallurgical processes aims for the reduction of between-user variability. To ensure conservative estimates, the description of the basic assumptions was improved (especially in “powder handling,” e.g., including a stratification for “Cleaning & Maintenance”, PROC 28). The database for non-volatile liquids was extended.

For all adjustments, the model developers tried to keep the parts of the program validated as “good” as unchanged as possible.

In summary, it can be concluded that the affected tool developers took action and considered the results of the ETEAM project for their revisions and updates of the tools. It remains to be evaluated whether the updates and research activities have the desired effects with regard to conservatism of exposure estimation and reduction of the variability between different users. For this evaluation, additional validation exercises are necessary. It is to be expected that validation of exposure assessment tools is a permanent activity that should be performed by all concerned stakeholders.

## 8. Authorities Using Models for Regulatory Assessments

Tier 1 exposure tools are preferably used in REACH registrations but also in the other REACH processes like Substances of Very High Concern (SVHC) identification, proposals for restriction, substance evaluation and authorisation. In these processes, REACH authorities play different roles which include in some cases also exposure assessments. When exposure assessments are performed or evaluated, mostly ECHA and Member State Competent Authorities for REACH (MSCA) as regulators have this responsibility in the different REACH processes. One of the interesting results of the BURE in the ETEAM project was that the quality of the exposure assessments performed with the different tools does not correlate to the position of the individual. Therefore, obviously, regulators are as prone to the same issues of validity as any other user.

### 8.1. European Chemicals Agency (ECHA)

During the last years, ECHA changed the process of compliance checks in dossier evaluations. Over the years, different toxicological endpoints were in the focus of the dossier evaluation. For some so-called “full compliance checks,” also exposure and use-related evaluations were performed by ECHA and concluded in decisions that challenged the use of exposure tools in specific cases. The ECHA decisions, for example, concluded that exposure assessments were performed outside the boundaries of the used tool and needed updates. Additionally, ECHA identified “REACH compliance” as an agency priority for 2019 [68].

ECHA’s Guidance on Information Requirements and Chemical Safety Assessment Chapter R.14: Occupational exposure assessment (short R14) was updated in 2016 [1]. Next to other issues, the current R14 version has now validation information integrated. For each of the tools evaluated in the ETEAM project, a sub-chapter was added that describes the status of validation and by that enables tool users to decide to some extent about this aspect of tool performance. However, R14 has not been updated since 2016. Therefore, some of the validation exercises performed in the last years have not been taken into account and the latest versions of some of tools are not described in the guidance.

### 8.2. Member State Competent Authorities for REACH (MSCA)

During substance evaluation—one of the important tasks for MSCAs in the framework of REACH—also exposure and use related information needs to be assessed by MSCA. Next to other sources, the registration dossiers are important for this. Therefore, the correct use of exposure assessment tools can be relevant in substance evaluation (SEv). 

When the evaluating MSCA cannot find exposure and use-related information of higher quality, modelling will be relevant for SEv and the validity of exposure assessment tools can become vital for regulatory decisions. Having in mind the limitations of tier 1 worker exposure tools as described above, MSCA might use these tools with caution before preparing regulatory RMMs like SVHC identification or restriction proposals. When preparing these kinds of dossiers based on modelled exposure estimates, especially those PROCs or activities that were identified as critical (see Table 2), scenarios associated with greater between-user variability and the effectiveness of RMMs at workplaces (e.g., LEV, general ventilation, dustiness) shall be thoroughly evaluated. Ultimately, it should be kept in mind by MSCA that REACH, for instance, offers a regulatory basis to request independent measurement data by way of the SEv process. The aim of the substance evaluation process is to clarify whether substances under evaluation may pose a risk and need additional regulation.

### 8.3. REACH Exposure Expert Group (REEG)

The REEG is an informal group of experts from authorities (Member States and ECHA) addressing the uses of chemicals and the related human and environmental exposure in the context of REACH. Its purpose is to enhance among experts the discussions, collaboration and coordination of activities on use- and exposure-related aspects to ensure an adequate link to the activities of the Risk Management and Evaluation Platform (RiME+) and to reach common understandings. The REEG programme for 2020 includes an action point 2.1 that aims at the consolidation of the different worker exposure estimation tools applied under REACH into a common framework (including communication aspects). Within this action, points on the limitations in reliability of exposure estimates (use situations, substance properties, tool-specific strength and weaknesses) are also addressed. This task is strongly connected to the ENES action 3.2 which is described below.

## 9. Consequences Drawn by Industry

For worker exposure assessments, different stakeholders from industry play different roles. Obviously individual companies from industry submit registrations to ECHA and use tier 1 tools for this purpose, but industry also has a role as associations for sectors of manufacturing and users of chemicals. These associations try to bundle information and provide sector-specific guidance and information.

### 9.1. Industry Associations

As communication in the supply chain (e.g., exposure scenarios) was identified early in the REACH processes as a major issue for improvement, industry associations for years now have taken a number of activities to improve and develop tools for supply chain communication. Two of these activities shall be described here.

Cefic, Concawe, Eurometaux, Fecc, and the Downstream Users of Chemicals Coordination Group (DUCC) together with ECHA started the “Exchange Network on Exposure Scenarios” (ENES, https://echa.europa.eu/about-us/exchange-network-on-exposure-scenarios) [69]. This collaborative network aims to identify good practices on preparing and implementing exposure scenarios, and to develop effective communication between actors in the supply chain. Member State competent authorities are active participants in ENES as are several downstream end-user sector organisations.

Within the ENES Work Programme 2020, next to other activities is the action 3.2 to “consolidate the different worker exposure tools into a common framework” that was started early 2018. This activity has short-term and long-term goals. The short-term goals are:Identification of potential platform, partners and process for the consolidation process;Workshops to take stock on what has been done in field during the last years;Analysis where existing worker exposure assessment the tools overlap and where they complement each other (based on SECO and ETEAM work);Identify weaknesses that may lead to significantly wrong assessments;Agree on a consolidation and update plan with involvement from member states and tool owners.

The long-term goals are:More transparent assessments;Less challenges by authorities (e.g., in context of SEv and authorisation);More consistent communication down the supply chain.

However, this ENES action is not yet finished and therefore it remains to be evaluated in the future how successful this activity will be and how the expected results can be implemented, e.g., in guidance documents, exposure assessment tools, and finally, in regulatory exposure assessments.

The second activity mainly driven by industry associations and presented here—as it was considered a reasonable idea to follow by BURE participants—are the use maps (https://echa.europa.eu/csr-es-roadmap/use-maps/concept) [70]. Use maps collect sector-specific exposure and use information and present it in a REACH-exposure-scenario-conform way for consumer, environmental and worker exposure assessments. The specific worker exposure descriptions (SWED) inform on OC and RMM for activities by workers. Registrants can use the information as an input to their exposure assessments. SWEDs include suitable standard phrases that help registrants effectively communicate the exposure scenarios attached to the safety data sheet to the downstream user. Table 3 presents the list of currently available use maps.

By using this information for exposure assessments for REACH purposes, a significant amount of variations between users of worker exposure assessment tools should be avoided. However, the SWEDS are not underpinned by independent measurement data yet and in registration dossiers (see below), the effect of this improvement cannot be yet observed.

Next to activities aiming for improvements in the supply chain communication, some industry sectors also developed their own tailored exposure models as one way to harmonise exposure assessment. The European Solvents Industry Group (ESIG), for example, developed the Generic Exposure Scenario (GES) Risk and Exposure Tool (EGRET) [71]. The MEASE tool, as a second example, was developed specifically for the estimation of occupational inhalation and dermal exposure to metals and inorganic substances on behalf of Eurometaux.

### 9.2. Registrants

A number of activities have been started by the European Commission, ECHA, MSCA and industry to evaluate the status of REACH registrations. The question asked how often—if at all—REACH registrations were updated by the affected registrants. In general, it was observed that only a limited number of dossiers have been updated so far and that industry is less than proactive in this regard. Specifically for worker exposure assessments, the authors (working for an authority with access to registration dossiers) could not identify improvements in the dossiers so far. No changes in the chemical safety assessments of REACH registrations were made because of the new knowledge about the validity of exposure assessment tools. Neither the findings of the ETEAM project or other studies nor new versions the tools resulted in updates of the registrations. This observation seems to hold true in all REACH processes such as registration, dossier and substance evaluation, but also restriction and authorisation. However, the issue of data quality and dossier improvement received some attention by industry [72] because registrants have obligations according to article 22 of REACH. Article 22 of REACH obliges the registrants to update the registration dossier “without undue delay” for several reasons. One reason is new information about risks caused by the use of a substance that leads to changes of the chemical safety report or the safety data sheet. Therefore, registrants should examine whether their assessments in their chemical safety reports are still valid in the light of the validation studies regarding exposure assessment tools. Registrants should especially pay attention to uncertain assessments because of e.g., specific activities/substance property combinations or use of RMM if the Risk Characterization Ratio (RCR) was already too close to one.

## 10. Additional Necessary Consequences

A number of additional activities to improve regulatory exposure assessments seem possible and necessary by different stakeholders. The already stated, above-mentioned activities should be continued, intensified and evaluated regarding their efficiency and effectiveness.

For the future activities, the authors of this publication (representing the opinion of the unit responsible for occupational exposure assessment in the German CA) would prioritize the following activities by the different stakeholders.

### 10.1. Registrants, Post-Registration Phase of REACH

As all phase-in substances are now registered under REACH and no further registration deadlines are consuming registrant’s resources, it is now time to validate the exposure scenarios already submitted in registration dossiers. Registrants should especially take into account the following aspects of exposure assessments: Was the used worker exposure tool applied within the applicability domain or was the tool “stretched” to its limits? Are the uses in the registration affected by the “critical PROCs” or by activities with high exposure potential? Was the effectiveness of RMMs (LEV, general ventilation, Personal Protective Equipment (PPE)) critically considered or was it overrated or inadequately used for exposure reduction? Were the morphology and the physical state of the substance adequately accounted for, keeping in mind that exposure assessment of dusty substances is more problematic for most tools than liquids? How was the quality of the exposure assessments assured? Is there room for improvement, e.g., application of different models, team coding, more training for assessors? It may be appropriate to consider the most conservative estimate if the exposure assessment is done on a tier 1 level. Is the risk really adequately controlled, keeping in mind the uncertainty of the exposure assessment in question? This might be relevant if the RCR is below but close to one. Or, is a higher tier assessment necessary/reasonable? Are other means of exposure assessments than tier 1 tool modelling available (e.g., read across to analogous exposure measurements, specific measurements at downstream user sites)?

An RCR below 1 is assumed to indicate that risks are sufficiently controlled, thus making higher-tier estimation unnecessary. However, the uncertainty and potentially insufficient conservatism of tier 1 exposure estimates may make RCRs below but close to 1 questionable [46]. In addition, the between-user reliability study (BURE) results suggest that when presented with brief, identical descriptions of exposure situations, user variation in the choice of input parameters can lead to very different results. As a consequence of the between-user variation and the model uncertainty, more confidence in the level of conservatism may be necessary. This should be based on insights provided by the evaluation of independent measurement data for the exposure situation under consideration.

### 10.2. ECHA and other Authorities

ECHA and other authorities (e.g., the European Commission, MSCA) are supposed to give guidance to industry and provide help to difficult questions. This is usually done by guidance documents, helpdesks and workshops. However, some of the most pressing questions resulting from tool evaluations are not yet addressed in the relevant guidance and need to be discussed not only scientifically but also on a political level. The authors of this publication would prioritise the following items by authorities:

The degree of conservatism that should be implemented in any exposure assessment has not been defined and should be a matter of debate of regulators. The following questions regarding conservatism of exposure assessment are not to be answered only scientifically:

How can conservatism of exposure assessment tools be measured? How high should it be for a tier 1 assessment? How can conservatism be defined for tier 1 tools? How conservative should higher tier assessments be?

The above-drafted questions also have a political dimension and cannot be answered scientifically or in technical guidance alone. However, different authors in the above-referenced studies developed proposals with regard to the conservatism of exposure assessment.

Additionally, it is important to note that first tier RCRs below but close to one may be questionable in cases of high uncertainty. The level of conservatism may not be sufficient; thus a more accurate, less uncertain assessment may still be necessary. In these cases, not only the exposure part of a risk assessment shall be evaluated and improved in case of doubt, preferably by providing quality assured measurements. The hazards assessment could also be refined in many cases. Potentially additional animal testing is a prerequisite for improved hazard assessment. Therefore, refining of the exposure assessment should be preferable. Obviously, this again has also a political dimension.

Next to these general issues, also the more technical aspects of worker exposure assessments should be addressed by ECHA, and this can be done in the next edition of the relevant guidance. The most relevant outcomes of the ETEAM project in this regard are:Assessment of and guidance on how to deal with “critical” workplace activities, tasks and scenarios (e.g., to be categorized as use descriptors like PROCs);Assessment of the efficiency of RMMs at workplaces (e.g., LEV, PPE);Guidance on how to deal correctly with the morphology of a substance in exposure assessments;Guidance on how to deal with quality management for exposure modelling.

### 10.3. Training of Tool Users

User training for exposure models and education of exposure assessors, respectively, can and should be improved. However, the means for this kind of improvements are to be developed. 

While great weight is given to the training and competence for workplace exposure measurements or toxicologists carrying out or evaluating toxicological studies, there is no similar requirement for users when generating and interpreting results from exposure assessment tools. As these modelled estimates are frequently used instead of workplace measurements, this absence of quality assurance seems somewhat contradictory. The same holds true for the available tools. Whereas calibrated sampling equipment and laboratory accreditation for chemical analyses are considered essential in ensuring the quality and reliability of measurement data, no analogous methods of quality control are applied to exposure assessment tools.

There is therefore, on the one hand, a responsibility of tool users to make sure that they are competent and able to perform the task. On the other hand, the tools need quality assurance as described above. As also described above, the tool owners have already reacted in this regard. A similar development for tool users is not observable so far. In order to support tool users in meeting their responsibility, the following proposals could be a way forward: Round-Robin-like quality control mechanisms that compare assessment results of different assessors could—in the long run—reduce between-user variations. The installation of certified trainings for exposure assessors, especially where exposure assessments will be used for regulatory decisions, will improve the quality of training for exposure scientists and the quality of regulatory decisions. This should be supported by the development and establishment of fit-for-purpose training courses for regulatory exposure assessments. Team assessment for critical exposure assessments (e.g., when regulatory action like authorisation or restriction is planned) could be a way forward. Previous studies postulated that team assessments and consensus finding can increase the validity of subjective assessments [53,73,74].

Obviously, the problem of these proposals is that they cannot be realised by tool users or single companies. Even authorities will not succeed when acting alone. A joint effort is necessary here by registrants, industry associations, regulators (European and member state level) and scientists (e.g., tool developers, university, scientific institutes,…). The involvement of the scientific community (i.e., relevant scientific societies) as well as an independent body for quality assurance is obligatory.

### 10.4. Tools

Further tool developments and improvements as planned by several of the tool developers should consider the following issues:Improvement with regard to user friendliness and help systems;Guidance for users to choose the correct input parameters.

The ETEAM project has shown that there are some parameters which are prone to induce a high level of variability due to their vague or insufficiently detailed definition. In particular, these are the: use categorisation for all tools;intrinsic dustiness which is defined qualitatively;type of setting (professional/industrial), and;definition of RMM.

## 11. Conclusions

Considering the results of all of the validation studies—especially the different work packages in the ETEAM project, the investigations by Landberg et al., the developments by Savic et al. and many others—dealing with different aspects of model validity (i.e., conceptual evaluation, uncertainty analysis, BURE, external validation, stratification) a number of positive features of the exposure assessment tools have been identified. Tier 1 tools are generally perceived to be easy to access, install, use and understand, and appear in most cases to overestimate exposure. However, there are also a number of aspects of the tools which were associated with underestimation of exposure. Especially, different studies identified ECETOC TRA (v2 and v3) and EMKG-EXPO-TOOL to underestimate different exposure situations. With regard to this issue, some suggestions are made especially by Lamb et al., but also by other working groups and in this publication regarding the following main areas of development:Definition of the expected level of conservatism;Quality control for exposure modelling;Support networks for exposure information;Improvement of user competency;Background information of and guidance for the tools;Clarity for the correct selection of the domain (professional/industrial);PROC code/activity descriptor;Definition of RMM, and;Influence of the morphology of assessed substances (dust).

Since all of the evaluated tools in the ETEAM project are classified as screening tools, it cannot be expected that all details of the OC and RMM are sufficiently specified for their subsequent practical implementation in workplaces at downstream user sites. Therefore, it is still necessary for the downstream user to interpret the provided scenario in a correct way and to perform an additional translation step that enables implementation of such model parameters (e.g., LEV efficiency) on a practical level. It has to be noted that the ETEAM project and all the other studies did not explicitly address operational aspects related to the practical implementation of the models. It is therefore recommended that future studies should do so. 

Implementation of the suggested improvements, both in isolation and combination, could help to improve the validity of tool exposure assessments and reduce inappropriate and inconsistent application of the tools. This could ensure the protection of worker health and avoid unnecessary RMM.

## Figures and Tables

**Table 1 ijerph-17-04589-t001:** Compilation of study reports and publications of the Evaluation of the Tier 1 Exposure Assessment Models (ETEAM) project.

ETEAM Sub-Project	Publication
Conceptual evaluation and uncertainty analysis	Sub-Study Report on Gathering of Background Information and Conceptual Evaluation [50] Sub-Study Report on Gathering of Background Information and Conceptual Evaluation [51]
Operational analysis	Sub-Study Report on User-Friendliness of Tier 1 Exposure Assessment Tools under REACH [52] Sub-Study Report on Between-User Reliability Exercise (BURE) and Workshop [53] Between-User Reliability of Tier 1 Exposure Assessment Tools Used Under REACH [45]
External validation	Sub-Study Report on External Validation Exercise [54] Validation of Lower Tier Exposure Tools Used for REACH: Comparison of Tools Estimates with Available Exposure Measurements [47] Further Stratification of the ETEAM Study Results [55]

**Table 2 ijerph-17-04589-t002:** Critical PROCS and influence of LEV in ECETOC TRAv3, EMKG-EXPO-TOOL, STOFFENMANAGER^®^.

Tool	Overall Conclusions on	
	Volatile Liquids	Powders
ECETOC TRAv3	medium to high levels of conservatism for PROC 4, 8a, 8b, 9, 10, 11, 13	high levels of conservatism in the majority of cases
	low levels of conservatism for PROC 5, 7 14, 19	low levels of conservatism for PROC 8a, 14
	overestimation of LEV efficiency for PROC 7, 8a, 10, 13, 14, 19	influence of LEV efficiency is not clear in all cases
EMKG-EXPO-TOOL	medium to high levels of conservatism for all cases	low levels of conservatism for PROC 5, 8, 14
		influence of LEV efficiency is not clear
STOFFENMANAGER^®^ 4.5	medium to high levels of conservatism for all cases, except for PROC 14	high levels of conservatism for all cases, except for PROC 8a

**Table 3 ijerph-17-04589-t003:** Use Maps as available on the European Chemicals Agency (ECHA) web site.

Sector Association	Sector	Covered Uses/Products
AISE—International Association for Soaps, Detergents and Maintenance Products	Soaps, Detergents and Maintenance products	industrial, professional (and consumer) (end)uses of detergent products
Concawe	Fuels	Fuels uses
Cosmetics Europe	Cosmetics and personal care products	Formulation, professional use (and consumer use) of cosmetic products.
ECPA—European Crop Protection Association	Plant protection products	Professional use and consumer use of plant protection products (spray application, granular application including treated seeds)
EFCC—European Federation for Construction Chemicals	Construction Chemicals	Formulation of construction chemicals, professional use of construction chemicals, e.g., resins, hardeners, organic binders, additives, inorganic binders
ESIG—European Solvents Industry Group	End-products containing solvents	uses of solvents across the majority of their applications for Industrial Sites, Professional Workers and Consumers,
EuPC—European Plastics Converters	Plastics Additives	Production of plastisol, masterbatches and compounds, and the production of Plastics Articles.
FEICA—Association of the European Adhesive and Sealant Industry	adhesives and sealants	Industrial uses, Professional uses, Consumer uses of several types of substances—e.g., solvents, additives, fillers, catalysts, … in adhesives and sealants
Fertilizers Europe	Fertilisers products	Formulation, industrial, professional and consumer uses of fertilisers products
I&P Europe/I&P Europe Imaging and printing products	Imaging and printing products	Formulation, industrial/professional and consumer use of chemicals from the imaging and printing industry, e.g., pressroom chemicals

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
