# Peer review of "Validity of Tier 1 Modelling Tools and Impacts on Exposure Assessments within REACH Registrations—ETEAM Project, Validation Studies and Consequences"

_ijerph, 2020, doi:10.3390/ijerph17124589_

Round 1

Reviewer 1 Report

General comments

  • This paper summarizes in one place the most common issues identified with the common tier 1 models across the various validation/evaluation papers, which is a useful exercise and will hopefully provide a convenient reference for the risk assessor.
  • Please take care when summarizing papers that reusing sentences/significant fragments from abstracts should be avoided.
  • The first half of the paper could be shortened and made more concise focusing more closely on any identified issues (sections up to “discussion of major results”) for follow up with recommendations in the latter section of the paper, and description of the response in recent years to the identified issues.
  • The authors should decide whether to include ART (as a tier 2 model) or not, given the focus is tier 1 only. This would help shorten and focus the paper.
  • While the paper understandably focuses on inhalation exposure, the biggest gap continues to be around dermal exposure assessments, and this should not be set so lightly set aside, particularly in a paper wishing to focus future developments (e.g. reference page 2, 50-51).
  • While aiming to be broader in scope than an ETEAM paper (page 2, 46-47, 60-63), the analysis/recommendations never-the-less focus primarily on ETEAM. The authors should consider simplifying and focusing just on ETEAM results, helping with length of the paper, or paying more attention to incorporate observations from the other research groups into the recommendations in the latter part of the paper.
  • The use of bullet point lists is used excessively and should be shortened to more concise in-line text using the author’s own wording. Please correct throughout the text.
  • Care should be taken in referring to PROCs as an exposure determinant: this has been hard coded in ECETOC TRA as an input parameter. PROCs are fundamentally a use descriptor (any reference to R12 is also missing), with no absolute definition: they are just a label to aid communication. Different models can hence legitimately calculate different exposures all while using the same PROC as a label (e.g. TREXMO etc has tried to draw equivalence between models for a given conceptual PROC). For example, the applicability domain for a given PROC in ECETOC TRA (which is actually determined by the underlying data) did not change with the refined PROC descriptions changed in the R12 guidance update.
  • PROCs should be defined somewhere in the text/footnote, as the average reader will (unfortunately) not know them by heart.
  • Pay attention to reference style/accuracy.

Specific comments

Page 2, 39: The tier 1 tools are designed to give a conservative exposure estimate based on a limited number of exposure determinants.

Page 2, 42-44: Or refine the hazard e.g. through further testing.

Page 2, 55-56: Excluding dermal exposure from the scope of the paper reduces its usefulness.

Page 2, 69: Perhaps worth noting these are all papers on EASE.

Page 3, 98: Include reference to R12 guidance.

Page 3, 103-104: Rather than generalise, the conditions under which underestimation was observed should be summarised: e.g. high VP liquid and for which PROCs.

Page 3, 123: That the group is American is not relevant.

Page 4, 140-141, 152-153: Pay attention to correct referencing of authors, and consider a less familiar style.

Page 4, 176-178: It should be noted that use of ECETOC TRA for assessing pesticide active substances is explicitly put out-of-scope in the ECETOC TRA guidance, nor is its use accepted by any pesticide regulatory authority: this was an unfortunate misuse of the tool.

Page 4, 179-181: Please be more precise as to which combination of scenarios and substance properties were found to be underestimated.

Page 5, 183: That the group was Japanese is not relevant.

Page 5, 187-189: Please be more precise as to which combination of substance properties and PROCs were underestimated.

Page 5, 196-197: Which PROCs were assessed.

Page 5, 200: The tool did not result in unsafe exposure, needs rewording.

Page 5, 212-214: That Cherrie et al’s explanation was not satisfactory is the author’s opinion, and should be more adequately substantiated.

Page 5, 215: That the group is Korean is not relevant.

Page 5, 223-225: It would be helpful for the later discussion to identify which PROCs and under what conditions had issues.

Page 5, 229-230: Identify which PROCs were at issue.

Page 6, 245-246: While a comprehensive overview, the conclusions from Spinazze et al should be treated with some caution. PROCs are hard coded into ECETOC TRA, therefore it is dangerous to categorically claim that Stoffenmanager/ART do or don’t cover a given PROC without the detailed contextual information at hand. Spinazze et al were not immune from the inter-user variability (highlighted by the authors) in their PROC assignments, based only on their paper based exercise.

Page 7, 303: Comma between BAUA and ECHA.

Page 7, 308-316: Text is unclear if it refers to the situation pre- or post- ETEAM, as hopefully the project improved the situation.

Page 7, 310: Intended meaning of “characterized by input parameters” is not clear: all models have input parameters.

Page 7, 315-316: ETEAM lead to questions or also provided some answers?

Page 8, 343: Table 1 not referenced from the text. It appears that subsequent headings are linked to this table, but this should be made clearer.

Page 8, 352-354: Very important advice well worth emphasising in the conclusions.

Page 8, 355: The authors should consider making references to other researchers who already identified inter-user variability as a significant source of variation:

  • Money A, Robinson C, Agius R, de Vocht F (2016) Wishful Thinking? Inside the Black Box of Exposure Assessment. Ann Occup Hyg; 60(4): 421–431.
  • Savic N, Lee EG, Gasic B, Vernez D (2019) Inter-assessor Agreement for TREXMO and Its Models Outside the Translation Framework. Ann Work Exp Health; 63(7): 814-820.
  • Riedmann RA, Gasic B, Vernez D (2015) Sensitivity analysis, dominant factors, and robustness of the ECETOC TRA v3, Stoffenmanager 4.5, and ART 1.5 occupational exposure models. Risk Analysis; 35(2): 211-225.
  • Savic N, Racordon D, Buchs D, Gasic B, Vernez D (2016) TREXMO: A Translation Tool to Support the Use of Regulatory Occupational Exposure Models. Ann Occup Hyg; 60(8): 991-1008.
  • Schinkel J, Fransman W, McDonnell PE, Entink RK, Tielemans E, Kromhout H (2014) Reliability of the advanced REACH tool (ART). Ann Occup Hyg; 58(4): 450-468.

Page 8, 356: Define BURE in the text, and add reference to Lamb et al 2017.

Page 8, 364: Please note that R12 guidance was updated in December 2015 improving PROC descriptions and introducing new terms, and thus would presumably not be reflected in the study.

Page 9, 381: Is a reference missing here?

Page 9, 385: Reference to PROCs here is only relevant for ECETOC TRA model, and presumably it is meant the model is not conservative enough rather than the use descriptor.

Page 9, 391: The authors should also consider including conclusions drawn from other validation studies covered in the introduction sections.

Page 9, 394: “automated” use of tools was only associated with ECETOC TRA. BURE is not relevant here as the approach appeared to be to calculate for all PROCs, thus leaving it to the end user to select the correct one to comply with: over/under estimation would not be a natural outcome. In the reviewer’s opinion, more problematic with this approach is the potential lack of thought, and hence “plausibility check”, for reasonable RMM and OC e.g. full PPE/RPE & 5min duration for a task that should be full shift.

Page 9, 396: Or equally overestimation of exposure, with resulting needless RMM, which can also have a negative impact on worker compliance when it is really needed, costs, etc.

Page 9, 398-399, 402: Replace risk management measures with RMM.

Page 9, 402: Unnecessary animal testing in the pursuit of hazard refinement is also a potential outcome.

Page 9, 404: Replace “or” with “of”.

Page 9, 404: “It has been argued” – please attribute by whom. Should tier 1 tools not be conservative (i.e. overpredict for 90th percentile of cases), rather than be accurate?

Page 9, 418: Note the R12 guidance has been updated in December 2015, although ECETOC TRA descriptions possibly not. In the reviewer’s opinion, this has more to do with the user’s potential willingness to read guidance/manuals, and is thus partially related to the authors’ point about training.

Page 10, 431: Do the authors means some models can’t provide sufficiently conservative estimates for some activities?

Page 10, 436: High level of uncertainty in what?

Page 10, 440: Do the authors mean “…informed progression … in risk assessment … is an…”?

Page 10, 447-449: Text seems a bit whimsical. Suggest to reword.

Page 10, 461-463: It is worth noting that some industry sectors have developed their own tailored exposure models as one way to harmonise exposure assessment:

  • Zaleski RT, Qian H, Zelenka MP, George-Ares A, Money C (2014) European solvent industry group generic exposure scenario risk and exposure tool. J Expo Sci Environ Epidemiol; 24(1): 27-35
  • AISE, REACT Consumer Tool
  • EUROMETAUX, MEASE

Page 11, 489: Risk management measures already defined, use RMM. OC as well?

Page 11, 498: Is “van Tongeren M, Lamb J, Cherrie JW, MacCalman L, Basinas I, Hesse S (2017) Validation of lower tier exposure tools used for REACH: Comparison of tools estimates with available exposure measurements. Annals of Work Exposures and Health, 2017.” a better reference here?

Page 11, 502-504: Perhaps the authors wish to make nuanced comment on the merits of funded/resourced software development versus “free” software.

Page 12, 533: Referencing an email in this context may not be appropriate.

Page 12, 558: This appears to be a section heading, and it would help readability if it was differentiated somewhat from the following subtitles.

Page 12, 562-564: It is worth pointing out that as users of the same exposure assessment software, regulators are as prone to the same issues as any other e.g. between user variability, etc.

Page 13, 584: Define SEv for first time.

Page 13, 586: Replace “get” with “be”.

Page 13, 590: Presumably reference to Table 2.

Page 13, 590-592: Those PROCs, or activities (thus making it more generic comment than ECETOC TRA), also those associated with greater between user variability.

Page 14, 612: Replace producers with manufacturers?

Page 14, 638: Use SEv.

Page 14, 649: Replace with OC and RMM.

Page 15, 652: Presumable reference to Table 3.

Page 15, 658: Replace with “yet observed.”

Page 15, 663: Replace with “…dossiers have been…”. Suggest replacing little with “less than”.

Page 15, 664: Specify from what no consequences have been noted, or replace with “improvements”. It should be made clear that this is the anecdotal observations from the authors as an authority with access (and hence insight) to the confidential CSR.

Page 15, 665-667: This sentence needs rewriting as the meaning is not clear.

Page 15, 674-675: Suggest to add an additional sentence making clear that it is not use of the exposure tool per se, but specific activities/substance property combinations which may lead to the need to rework those specific assessments. Changes to RMM would only be anticipated if the RCR was already too close to one (i.e. dependent on the tox profile/potency of the substance).

Page 16, 680-682: This sentence appears to contradict the declaration of publication (828-829).

Page 16, 685: Perhaps better wording would be to replace “threatening” with “consuming registrant’s resources”.

Page 16, 690: Suggest to make more generic by adding “or activities”.

Page 16, 719: Replace priories with prioritize.

Page 16, 723: “How can conservatism of tier 1 tools be defined?”

Page 17, 725: “How conservative should a tier 2 assessment be?” Note question on scope of this paper and tier 1 vs 2. Rather than assessment factors, should not an exposure estimate conservatism be defined in terms of percentile? E.g. 90th etc.

Page 17, 728: To some extent some answers should be available and agreed already from the long history of measurements for compliance with OELs, especially now that RAC have an expanded role.

Page 17, 735: “may still be necessary”.

Page 17, 736-737: “… may be evaluated and improved, but also the hazard assessment could be refined in many cases.” It may also be worth noting that potentially animal testing is a consequence, hence refining the exposure side of the equation should be preferable.

Page 17, 741: Suggest broadening from critical PROCs to include more generic activities.

Page 18, 780: Delete trailing “and”.

Page 18, 786: RMM

Page 18, 788: The original scope of the paper was understood to be broader than ETEAM project, and input from other research should be acknowledged.

Page 18, 795: A new reference should not be introduced in the conclusion.

Page 18, 802-805: Correct training “and”, and other punctuation.

Page 18, 807: OC and RMM.

Page 18, 816-817: RMM.

Page 21, 972: Check reference against journal style.

Author Response

Please find our comments in the attachment!

Reviewer 2 Report

The manuscript needs data tables, analyses etc. of the cited articles.  The title indicates it is focused on Tier I models. There are many mentions of Tier II models but for what reason? There are no analyses of either Tier I or Tier II data just a short discussion of each with no real conclusions or insights. The information that is provided is readily available from the abstracts of the cited literature.

Compare this draft manuscript to the excellent ETEAM final report on Tier I models published in 2015 and you will see what I mean.

https://www.baua.de/EN/Service/Publications/Report/F2303-D26-D28.pdf?__blob=publicationFile&v=4

There could be something here but in its current form does not add new information or innovative ways of looking at exposure models or data.

Author Response

Please find our comments in the attachment.

Round 2

Reviewer 1 Report

General comment: In my view still a bit too "bullet pointy", but as the authors state, this is somewhat a matter of style and taste. The changes that have been made in this area have generally improved the presentation and readability of the paper.

General comment: for the bullet point lists, for consistency decide on punctuation or not at the end of each sentence throughout the manuscript. e.g. delete or add “.” or “;” or “,” consistently for each point.

Comments below are mainly editorial corrections:

Page 2, line 61. “still less advanced than for inhalation exposure.”

Page 2, line 67. “validity has received a much greater attention”

Page 4, line 165. “companies and studying situations at the workplace.”

Page 4, line 165. Write “geometric mean”.

Page 4, line 168. “These last findings are published”

Page 5, line 189. Delete “)”.

Page 5, line 216. I didn’t explain the comment clearly enough, it was related to grammar. Presumably the activities modelled by EMKG-EXPO-TOOL were already taking place, hence the tool was not causally responsible for the over exposure. Suggest amending to “Estimation with the EMKG-EXPO-TOOL failed to identify unsafe exposure situations for all scenarios, which is in accordance with the measurement outcomes.”

Page 6, line 233. “Due to this lack of”

Page 10, line 424. Delete second occurrence of “always”.

Page 10, line 435. “information seems to not assist in reducing”

Page 10, line 445. Replace “model” with “exposure”.

Page 11, line 498-500. These two bullet points are hanging below the text with no explanation. Either add a sentence introducing them, or integrate into the preceding paragraph.

Page 13, line 599. Reference error.

Page 15, line 660. Reference error.

Page 17, line 737-8 “How conservative should higher tier assessments be?”

Page 18, line 775. “mechanisms that compare assessment results of different assessors could – in the long run – reduce”. That round-robin testing would reduce variability sounds like a reasonable assumption, but as it involves humans, might be a bit speculative!

Page 19, line 815. Delete “and” at the end of the sentence.

Page 19, line 816-18. Delete punctuation at the end of the sentences e.g. “.”, “;”.

Reviewer 2 Report

1) There is a large amount of text in the article that is unrelated to its stated purpose. Frequently Tier II model(s) are mentioned. Try leaving out discussions of Tier II models unless essential for some point being made regarding Tier I models.

2) Please reference the version of the model that was tested in the referenced article. Towards the end that is done more often but still not enough. This is important to the reader of the article when evaluating its impact on their REACH activities.

3) Starting on page 8 of 24 through to the beginning of the Conclusion paragraph (18 of 24) there is lengthy discussion of management, REACH registration processes, organizations (formal and informal), stakeholders etc. that add nothing to the goal/purpose of paper. Delete this section or as much as possible and save the scientifically DIRECTLY related sections.

The manuscript PDF was edited in acrobat. Many additional comments are included in document in "Comment boxes" and those boxes are linked to specific words/sentences in the manuscript.
